# Isolated proton bunch acceleration by a petawatt laser pulse

P. Hilz [1], T.M. Ostermayr [1,2], A. Huebl [3,4], V. Bagnoud[5,6], B. Borm[7], M. Bussmann [3], M. Gallei[8], J. Gebhard[1], D. Haffa[1], J. Hartmann[1], T. Kluge [3], F.H. Lindner[1], P. Neumayr[5], C.G. Schaefer[8], U. Schramm [3,4], P.G. Thirolf[1], T .F. Rösch[1], F. Wagner[5,6], B. Zielbauer[5] & J. Schreiber [1,2]

Often, the interpretation of experiments concerning the manipulation of the energy distribution of laser-accelerated ion bunches is complicated by the multitude of competing dynamic processes simultaneously contributing to recorded ion signals. Here we demonstrate experimentally the acceleration of a clean proton bunch. This was achieved with a microscopic and three-dimensionally confined near critical density plasma, which evolves from a 1 μm diameter plastic sphere, which is levitated and positioned with micrometer precision in the focus of a Petawatt laser pulse. The emitted proton bunch is reproducibly observed with central energies between 20 and 40 MeV and narrow energy spread (down to 25%) showing almost no low-energetic background. Together with three-dimensional particle-in-cell simulations we track the complete acceleration process, evidencing the transition from organized acceleration to Coulomb repulsion. This reveals limitations of current high power lasers and viable paths to optimize laser-driven ion sources.

[1] Fakultät für Physik, Ludwig-Maximilians-Universität München, München, Germany. [2] Max-Planck-Institut für Quantenoptik, Garching b. München, Germany. [3] Helmholtz-Zentrum Dresden - Rossendorf, Dresden, Germany. [4] Technische Universität Dresden, Dresden, Germany. [5] GSI Helmholtzzentrum für Schwerionenforschung, Darmstadt, Germany. [6] Helmholtz-Institut Jena, Jena, Germany. [7] Goethe Universität Frankfurt, Frankfurt, Germany. [8] Macromolecular Chemistry Department, Technische Universität Darmstadt, Darmstadt, Germany. Correspondence and requests for materials should be addressed to P.H. (email: Peter.Hilz@lmu.de)

Providing intense bursts of swift ions has gained particular interest[1–3] and proton kinetic energies exceeding 85 MeV have been recently demonstrated in various experiments[4,5]. The acceleration field is mediated via relativistic electrons, which induce MV/μm electric fields that vary in time and space. The correspondingly broad ion energy distributions could be narrowed by limiting the spatial extent of the ion reservoir on the surface of irradiated opaque foils[6,7] or by the use of droplets[8,9]. Reducing the foil thickness to the order of the laser skin depth also resulted in non-monotonic, peaked distributions with improved efficiency at higher ion energies[10,11]. Such radiation pressure or related volumetric acceleration mechanisms[12] have in common that the majority of electrons in the central part of the focus are coherently pushed by the light forces and then drag ions along. The experimentally observed ion signal, however, is typically blurred by superimposed ions from regions outside of this central part of the laser focus. The volumetric interaction of electrons with the laser field requires plasma densities around the critical density $n_c$. Plasma densities above $n_c$ are opaque for the laser light. For densities smaller than $n_c$ plasmas are transparent. By virtue of their 10 μm wavelength $CO_2$ lasers enable to access such realms with readily available gas targets[13,14]. The low density does however limit the accelerated particle numbers severely. Spatially limited targets, so-called mass-limited targets, have attracted severe interest in theory[15] and experiment[9,16,17].

In this work, we exploit a so far inaccessible target parameter space by irradiating isolated, levitated plastic spheres of 1 μm diameter with a PW laser of 1.054 μm wavelength. In the rising edge of the laser pulse, the sphere expands to a few micrometer-sized plasma of approximately critical density for the near-infrared laser pulse. The single proton bunch generated in the interaction contains a large fraction of all target protons and are accelerated to narrow energy bands down to 5 MeV full-width at half-maximum (FWHM). Central energies for consecutive laser shots range from 20 to 40 MeV.

## Results

**Experimental setup.** The laser pulses provided by the PHELIX PW laser at GSI (see Methods) contained 150 J energy with a pulse duration of 500 fs and were focused to a diameter of 3.7 ± 0.3 μm (FWHM of intensity). Plastic spheres of 1 μm diameter and hollow spheres with an outer diameter of 1 μm and 100 nm wall thickness served as targets. The microscopic target size inhibited any kind of mechanical support. Hence, we used a Paul

trap to levitate and position the targets. A direct opto-electronic feedback allowed us to position the targets with an accuracy of around ±1 μm in all three dimensions (see Methods). The dominant source of fluctuation in laser target overlap due to pointing jitter of the laser was minimized by placing the target 1 to 1.5 Rayleigh lengths out of focus, where the on-target peak laser intensity amounted up to $7 \times 10^{20}$ W/cm² (the peak intensity in best focus would have amounted to $2 \times 10^{21}$ W/cm²).

**Pre-plasma expansion.** The excellent nanosecond contrast of the PHELIX laser prevents premature target ionization and has been demonstrated in previous experiments with sub-micrometer thin foil targets[18]. The measured autocorrelation trace in Fig. 1a was scaled to our estimated peak intensity. Considering that the plasma formation is related to the laser-induced damage threshold that we measured on various thin, transparent plastic foils, we estimate the start of relevant plasma dynamics 200–100 ps prior to the peak. Instead of treating the following complex and complicated, three-dimensional pre-expansion dynamics theoretically, we can draw important conclusions on the plasma conditions during the main pulse interaction by evaluating the experimentally recorded intensity patterns in Fig. 1b–d. The observed circular fringes can be considered as a result of an automatically generated inline-holography[19], where the laser amplitude and phase in the center are disturbed by the target plasma and the outer regions are, depending on the size of plasma and laser focus, less or even not at all affected. We modeled the resulting intensity distributions at the plane of observation for different plasma dimensions (Fig. 1e–j) (see Methods). Under these simplified assumptions, the fringe pattern compares best with the experimental images when the peak electron density is close to the critical density. The transverse extent of this critical density plasma exhibits an FWHM of 5.5–7.0 μm. We note that although many factors (such as target position with respect to focal plane, asymmetric shape, non-Gaussian distribution function of plasma and laser) influence the observed diffraction pattern, our analysis supports that the general appearance, i.e. the number of visible fringes and their contrast, remains relatively constant over a large range of densities, in line with the observation (Fig. 1b–d) whenever we hit the target and successfully accelerated ions.

**Experimental results.** The emitted protons have been analyzed in laser propagation direction employing a magnetic slit

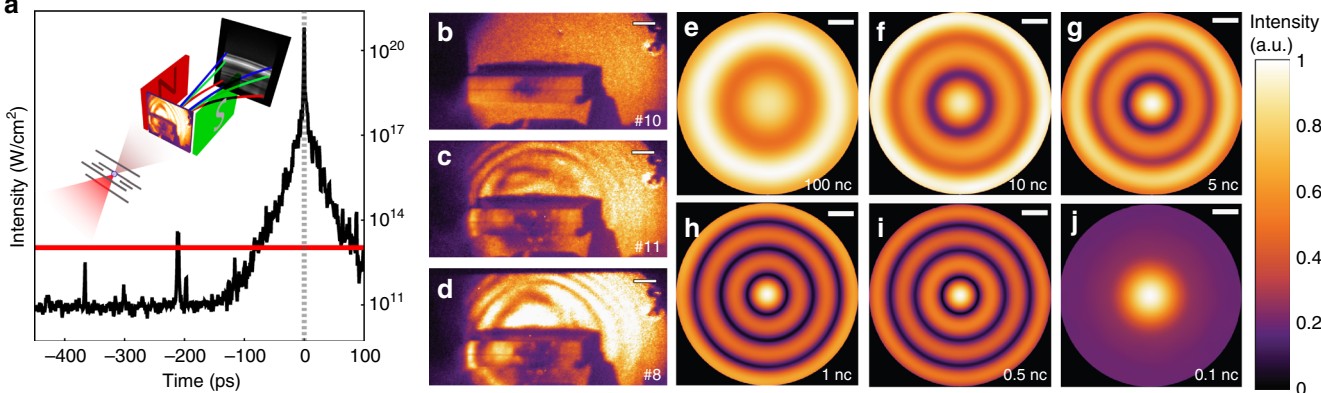

**Fig. 1** Pre-plasma formation. **a** Autocorrelation curve of the PHELIX laser pulse scaled to the maximum on-target intensity $7 \times 10^{20}$ W/cm². The horizontal line marks a typical value of plasma formation intensity for plastic. The inset shows the schematic experimental setup including the relevant diagnostics. Experimental recorded diffraction patterns at the scatter screen in front of the magnetic spectrometer for the case, **b** no target, **c** solid sphere, **d** hollow sphere are shown. The scale bar corresponds to 2 cm. **e–j** Simulated diffraction patterns for a variety of peak electron densities $n_m$ in units of critical density $n_c$ and corresponding plasma extension. The scale bar corresponds to 2 cm

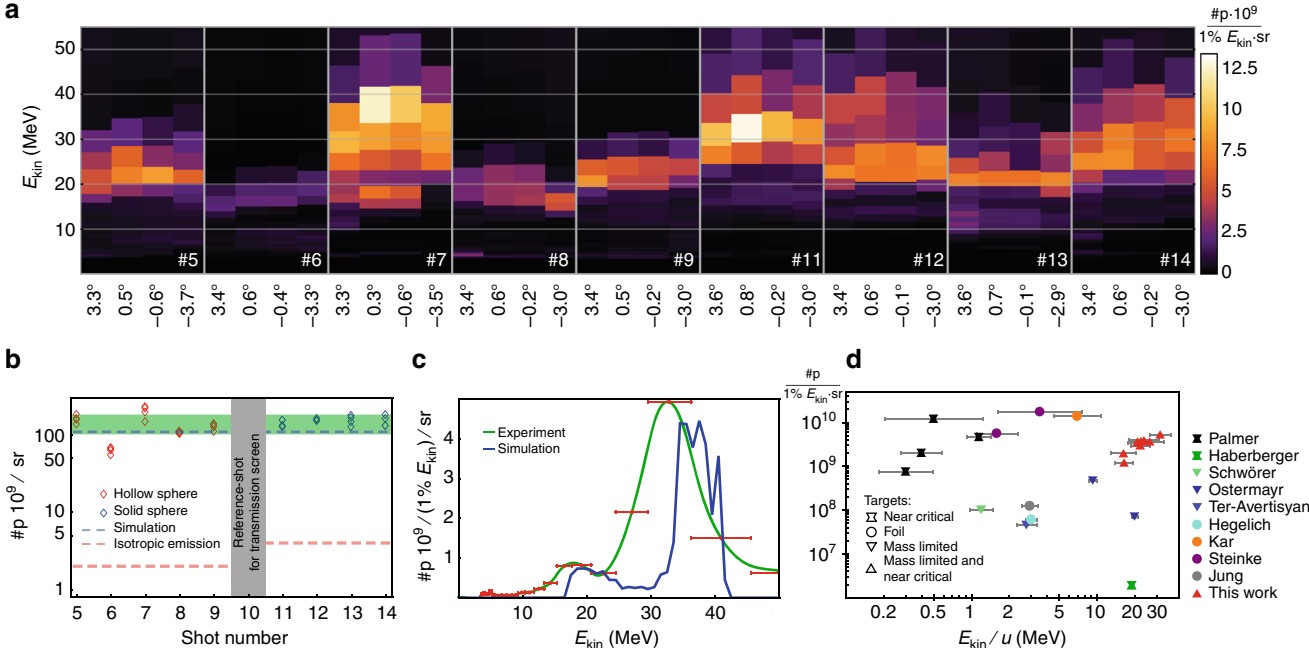

**Fig. 2** Experimental results. **a** Differential proton spectra for consecutive laser shots for various angles. **b** Proton number per solid angle from experiment compared to isotropic emission into $4\pi$ and numerical simulation, the green area represents the standard deviation of the experimental data. **c** Comparison of simulated differential proton spectrum and experiment (shot 11@0.8°), the red error bars indicate the spectrometer resolution. **d** Summary of differential ion yields for experiments reporting on narrow energy spread, represented by the horizontal error bars (FWHM). Cross-shaped symbols represent results obtained with $CO_2$-laser pulses in gaseous plasmas with critical density, circles represent results from thin foils, and triangles pointing down spherical solid targets (such as droplets). See Refs. [6-14]

spectrometer (see Methods), allowing single-shot angular resolved energy measurements and thus the absolute determination of differential proton spectra. In all laser shots, we measured proton kinetic energy distributions with narrow energy spreads. We observed no notable variation across the accessible angular range of $\pm 4°$, which we analyzed for four angles (Fig. 2a). Integrating the proton spectra over energy yields the number of protons per solid angle. For each shot, we evaluated the four angular spectra independently and plotted the results as diamond symbols in Fig. 2b. Solid and hollow spheres deliver similar results, despite the factor of two difference in particle numbers contained initially in the target. This supports our findings regarding target pre-expansion. Contrary to foil and mechanically mounted targets, the initial number of protons is known in our case. Herby we can deduce the #p/sr for an ideal Coulomb explosion, where all protons in the target are emitted into $4\pi$ sr (dashed red line in Fig. 2b). Our measured values for #p/sr are 30–100 times larger compared to the isotropic ideal Coulomb explosion. Though not visible in the small angular range of our particle spectrometer, this comparison evidences a large degree of directionality of the accelerated proton bunch, also in accordance with our simulations. In a follow-up experiment 2 years later we were able to reproduce the above findings.

**PIC simulation and comparison with experiment**. We performed 2D3V and 3D3V particle-in-cell simulations[20,21] to support our experimental results quantitatively and elucidate the underlying microscopic processes in more detail. It turned out that in the case of two-dimensional simulations we were not able to reproduce the experimental findings for a wide parameter range. Due to the high computational cost of three-dimensional simulations we were limited just to one single simulation. The initial conditions for plasma density and laser intensity

distribution are chosen in such a way to closely resemble the experimental ones (see Methods). In analogy to the experiments, we extract absolute differential proton spectra in forward direction (blue line in Fig. 2c) and compare it exemplarily to shot number #11 (green solid line) in Fig. 2c. Given the complexity of the involved physics and assumptions, the quantitative agreement of experimentally measured and simulated kinetic energy distribution is remarkably good, both in terms of energy and differential spectral amplitude. The number of protons per solid angle observed in simulation is represented by the blue dashed line in Fig. 2b and matches the experimentally determined values. Figure 2d compares our current results (red triangles) to recent experimental studies showing narrow energy band ion beams from laser–plasma interactions. The number of protons is normalized to an energy interval of 1% of the respective kinetic energy per nucleon and solid angle of 1 sr. Our results push the energy frontier of narrow band laser-driven ion sources with usable particle numbers. The spectral peak contains a factor of >30 more particles than earlier approaches reaching into comparable energy ranges. The limitations of achievable proton energies and possible routes for optimization for the presented approach unravel when studying the results of the three-dimensional PIC simulations in more details

## Discussion

Encouraged by the quantitative agreement of the differential spectrum in Fig. 2c, we elucidate its temporal evolution. Figure 3a reveals that the spectral distribution remains narrow throughout the complete interaction of the laser pulse with the plasma. Since the plasma is transparent to the laser from the beginning, the laser interacts with all electrons in a collective manner (Fig. 3b). In turn, the optical properties of the plasma act back on the laser and vice versa. The resulting normalized spatial intensity

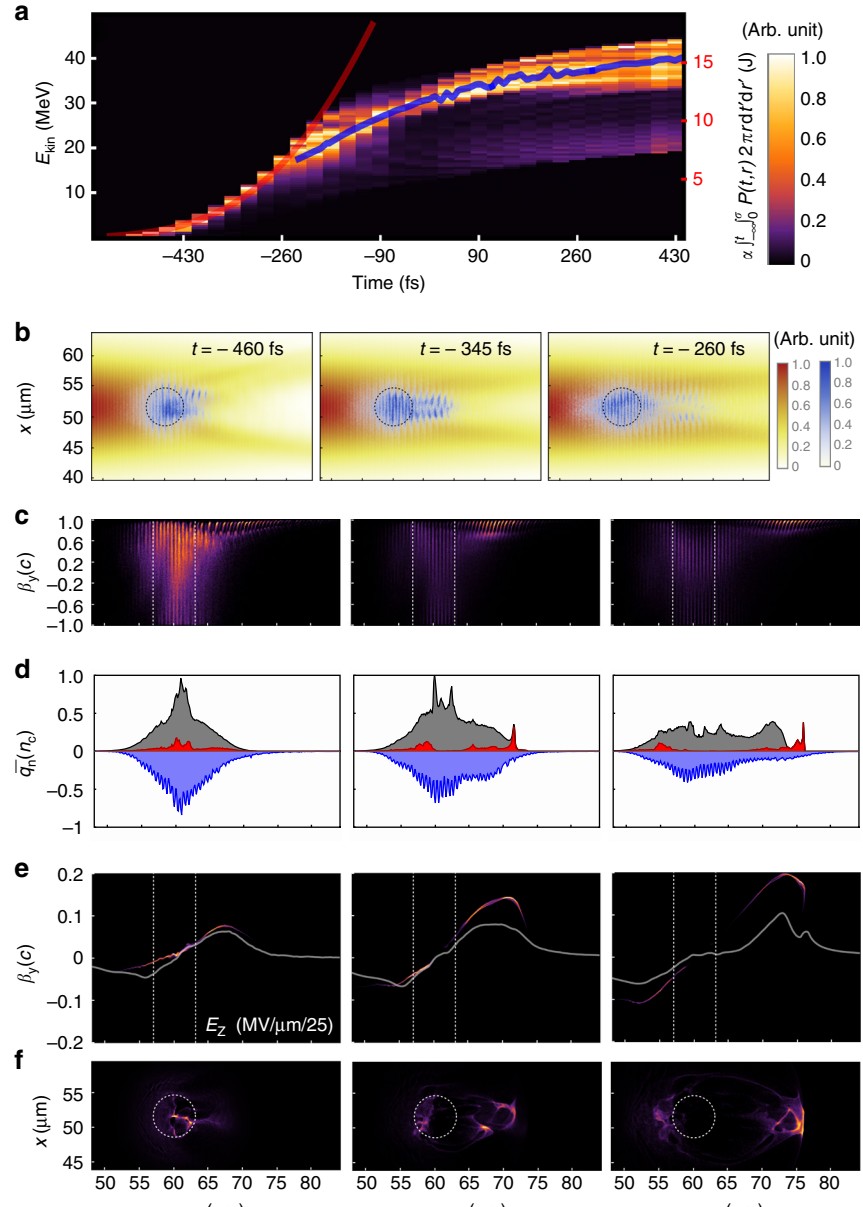

**Fig. 3** Particle-in-cell simulation and analysis. **a** Time evolution of proton energy distribution throughout the interaction with the laser pulse with power $P(t)$. The colormap represents the proton number/1% $E_{kin}$/sr normalized to a maximum of 1 for each individual time frame. The red curve represents the laser pulse energy that has passed the plasma $\alpha \int_{-\infty}^{t} \int_0^{\sigma} P(t,r)2\pi r dt' dr'$ with $\alpha = {\sim}2.4$ MeV/J ($\sigma = 3.0626$ μm), the blue curves correspond to estimates of free Coulomb repulsion from $t = -230$ fs onwards. **b** Individually normalized instantaneous laser intensity distribution and electron density distribution in the polarization plane. **c** Individually normalized longitudinal electron phase-space density. **d** Longitudinal on-axis charge density of all positive ions (gray), protons (red), and electrons (blue). **e** Individually normalized longitudinal phase space of protons superimposed to the accelerating field. **f** Individually normalized proton density distribution in the polarization plane of the laser. **a**, **c**, **e**, **f** use the same colormap

distribution in the polarization plane of the laser is shown for three time points (460, 345, and 260 fs before the peak intensity reaches the plasma) that are of interest for tracing the first phase of proton acceleration in Fig. 3b. The corresponding instantaneous on-axis laser intensity acting on the plasma amount to $6.8 \times 10^{19}$, $1.9 \times 10^{20}$, and $3.3 \times 10^{20}$ W/cm². During this period, the laser pulse infolds the plasma transversely, as visualized in the overlaid electron density distribution (blue colormap Fig. 2b). This is in close analogy to a metal sphere in the donut mode of an optical tweezer[22], only here the donut mode is self-induced by the optical properties of the plasma. Over time, a substantial fraction of the electrons leave the target and are accelerated to large

longitudinal velocities approaching the speed of light with growing intensity (Fig. 3c). As observable in their density distribution slice in Fig. 3b, they leave the indicated initial target region preferably in laser propagation direction. The laser intensity valley at the same time confines them transversely. In Fig. 3d we visualize the resulting charge imbalance via the longitudinal distribution of the mean charge densities obtained by transverse averaging the density slice of negative (electrons) and positive (ions and protons) particles over a range of 1.5 μm. Due to the limited number of available electrons and the asymmetric intensity distribution, the electron density distribution (blue filled curve) evolves asymmetrically, and so does the associated profile

of the electric field. In consequence ions are dragged preferably toward the intensity valley as well. Figure 3e evidences that the accelerating field distribution remains stable over an extended period of time, despite short-lived modulations observed in the charge densities. As protons are the ion species with a largest charge to mass ratio they are accelerated most promptly and develop a negative energy chirp in the monotonic slope of the field. When the fastest protons reach the peak of the field, which amounts to ~2.5 MV/μm, the tail of the bunch catches up, observable in the superimposed proton phase space. The narrowest energy spread is reached at $t = -50$ fs with 3.5% (1 MeV FWHM @ 29 MeV). It is interesting to note that the energy peak position of the detaching proton bunch closely follows the red line superimposed in Fig. 3a until $t = -260$ fs. It represents the cumulatively time and spatially integrated laser intensity, i.e. the laser energy that has passed through the plasma up until the respective time. The proportionality factor between the accumulated energy and central bunch energy is estimated in this case to 2.4 MeV/J. At $t = -260$ fs the bunch energy no longer follows this favorable scaling, which would otherwise result in a proton bunch with 163 MeV energy. Over the time the electron population in the target gets more and more depleted. Hence, the acceleration due to the electron jet is replaced gradually by Coulomb repulsion forces. Extracting the three-dimensional ion density distributions at time $t = -230$ fs, the further evolution of the proton bunch can be well described by considering pure Coulomb repulsion (see Methods). The time evolution of the proton energy distribution resulting from this calculation is visualized by the blue line in Fig. 3a. In the light of the above explanation, we draw one important conclusion. The initial acceleration phase, which only lasts until ~345 fs before the laser intensity peak, terminates due to strong electron heating and the depletion of electrons (Fig. 3b, c). The remaining time is dominated by Coulomb repulsion between the leading protons from carbon and oxygen ions. Nevertheless, the energy spread remains low. In order to increase the kinetic energy, the favorable scaling reflected by the red line in Fig. 3a suggests that we must seek to extend and maintain the well-controlled initial phase for as long as possible. In this context it is instructive to examine the conversion efficiency from laser energy into kinetic energy of protons/ions. In our simulation 58 J laser energy pass through the plasma. The accumulated final kinetic energy of all protons is 181 mJ, 196 mJ for oxygen ions, and 367 mJ for carbon ions, respectively. This leads to an overall energy conversion efficiency into protons of 0.3 and 1.3% into protons and ions combined. Until −260 fs only 6.3 J have passed the plasma, the remaining part of the laser pulse does not contribute to proton/ion acceleration. The effective laser energy conversion efficiency may therefore be arguably ~10× larger: 3.1% into protons and 11.8% for protons and ions combined. Consequently, future studies, both theoretical and experimental, should concentrate on spectral, temporal, polarization and spatial shaping of the laser pulse to further optimize the electron dynamics that is responsible for the generated acceleration fields in the first acceleration stage. On this track, target composition, density, and size represent further possibilities for optimization.

In conclusion, we note that, although not yet optimized by any means, our approach enables a target parameter space which in conjunction with a PW-class laser results in the reproducible emission of a proton bunch with high density and tens of MeV energies. Simulations show a total of $10^{10}$ protons in the generated bunch, equal to 14% of the initial reservoir (micro-sphere target) content. The simulations further suggest that the longitudinal proton bunch length is only 4 μm (Fig. 3f). Immediately behind the original target, the particle density reaches ~$10^{21}$ protons/cm$^3$. Proper particle optics may be able to recover those conditions to some extent at a place of interest for ultrahigh dose rate investigations as well as temporally and spatially resolved studies, but also direct collider experiments may be envisioned. The final transverse divergence full angle amounts to 20° with a flat distribution. This divergence establishes during the final Coulomb repulsion phase of the acceleration, while the proton bunch is even more directed during the initial acceleration phase. It is remarkable that a 500 fs long laser pulse can generate a short bunch of ~50 fs duration, four orders of magnitude smaller than conventional particle accelerators.

## Methods

**Detailed experimental setup.** Fig. 4 shows schematically our experimental setup. All numbered components are described in more detail in the following sections.

**PHELIX laser.** The experiment employed the PHELIX laser at GSI Helmholtzzentrum für Schwerionenforschung. PHELIX is a glass laser system based on chirped pulse amplification with a central wavelength of 1054 and 3 nm bandwidth. In the experiments the pulse energy amounted to 150 J with pulse duration of 500 fs. The laser pulse was focused by an 45° offaxis parabolic copper mirror with 400 mm focal length. The beam size amounted to 250 mm. The laser contrast was enhanced by a fast Pockels cell and optical parametric pulse cleaning techniques leading to an amplified spontaneous emission (ASE) level below $10^{11} \frac{W}{cm^2}$ with a duration of approximately 2.5 ns. The main pulse exhibits an exponential shoulder in which the intensity raises over 110 ps from ASE level to $10^{16} \frac{W}{cm^2}$. After this exponential slope the main pulse can be assumed to be Gaussian in shape. The laser is capable to deliver one shot every 90 min.

**Paul trap.** We employed a linear Paul trap for target positioning (Fig. 5). Axial confinement was achieved by end cap electrodes. For the employed targets typical trap voltages were 1000–2000 kV, with frequencies in the range of 800–1500 Hz for

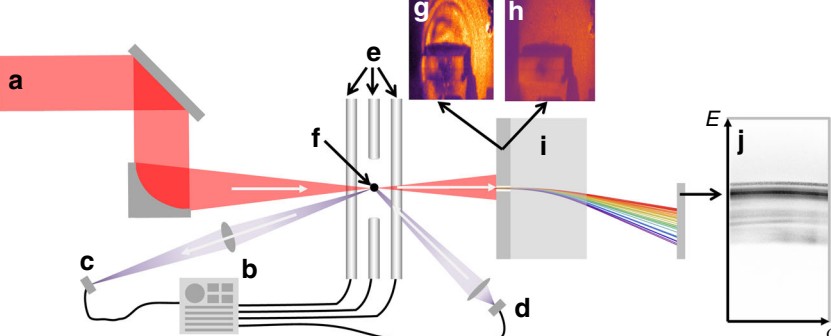

**Fig. 4** Schematic overview of the experimental setup. **a** Incoming PHELIX laser beam, **b** Paul trap power supply, **c** 660 nm illumination laser, **d** electro-optical diagnostic for target damping and positioning, **e** Paul trap electrodes, **f** trapped target scatter screen for transmitted light with **g** and without **h** target, **i** magnetic spectrometer, **j** IP raw proton/ion data (without degraders)

stable trap operation. Typical end cap voltages were 100–300 V (DC). The distance from the trap center to the quadrupole electrodes measured 8 mm. The distance between the endcaps amounted to 20 mm. Targets were extracted out of a reservoir by mechanical vibration. While falling into the already operating trap they got charged by an ion gun and subsequently captured. In a first step the amplitude of the trapped target was reduced by buffer gas cooling at $10^{-4}$ mbar. Final stabilization of the target position was achieved by electro-optical damping under vacuum conditions smaller than $10^{-5}$ mbar. Hereby, the target was illuminated by a 50 mW laser with 660 nm wavelength. Stray light from the particle was imaged by an tele-centric lens system onto a position sensitive diode (PSD). With a beam splitter a small fraction of this stray light was imaged onto a CCD camera. The PSD signal was used to damp the particle by superimposing additional fields on the trap electrodes. Overlap of the damped particle with the laser focus was ensured by the laser focus diagnostic. By the CCD signal we ensured that the target remained quiet and at the right position prior to a full system shot. Fifty milliseconds before irradiation of the target with the PHELIX laser mechanical shutters protected illumination laser, CCD, and PSD.

**Targets**. The PMMA hollow spheres were produced by covering polystyrene spheres (Fig. 6a) with a PMMA layer. The polystyrene cores were subsequently dissolved away (Fig. 6b). The solid PMMA spheres employed in the experiment were commercially available via microparticles GmbH.

**Simulation of diffraction pattern**. In the case of a sub-focus-sized mass-limited target, the laser is able to partially pass the target undisturbed. Regions containing plasma change phase and amplitude of the laser. We simulated diffraction patterns for various density distributions and compared it to experimentally recorded patterns. The laser was modeled by a Gaussian intensity distribution. To mimic divergence and focal spot size we used an $M^2$ of 2.8. Twenty micrometer after the focus a spherically symmetric electron density distribution $n(r) = n_\mathrm{m} \exp\left(-r^2/(2r_\sigma^2)\right)$ has been defined, where $n_\mathrm{m} = n_0 \cdot (9\pi/8)^{1/2} r_0^3/r_\sigma^3$ has been varied such that the total electron number was kept constant. The phase of the laser was modified according to the refractive index of a collisionless plasma $\eta(r) = (1 - n(r)/n_c)^{1/2}$ in regions where $n(r) < n_c$. For $n(r)_c$ the amplitude was set to 0. Figure 7 shows the assumed waveform aberrations for two different pre-expansion scenarios. The color scale indicates the amplitude of the wave. If the plasma is expanded to a degree where its density is 100% under-critical the plasma acts as a phase object. In the case of an smaller expansion the plasma is still over-critical in

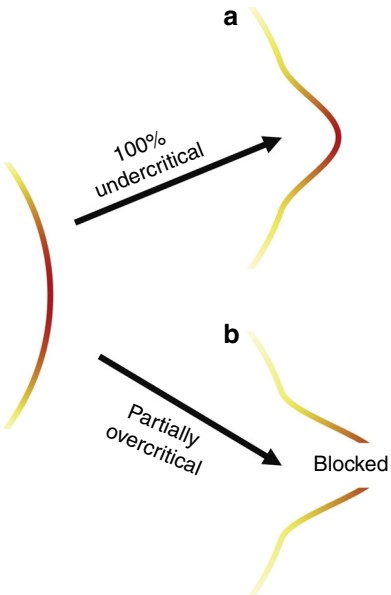

**Fig. 7** Schematic representation of waveform aberrations induced by the plasma. **a** In the case of a 100% under-critical plasma the phase of the wave is distorted, the wave amplitude remains unchanged. **b** Over-critical regions block the incoming wave

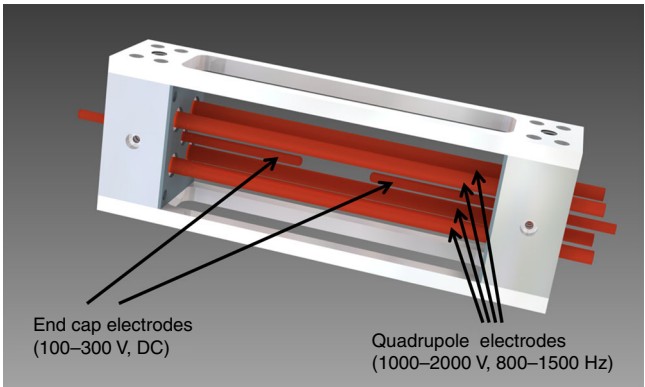

**Fig. 5** Linear Paul trap: 3D view of the employed Paul trap. Trap electrodes are shown in red. The distance between the end cap electrodes can be adjusted to account for different target sizes and materials (20 mm in the presented experiment)

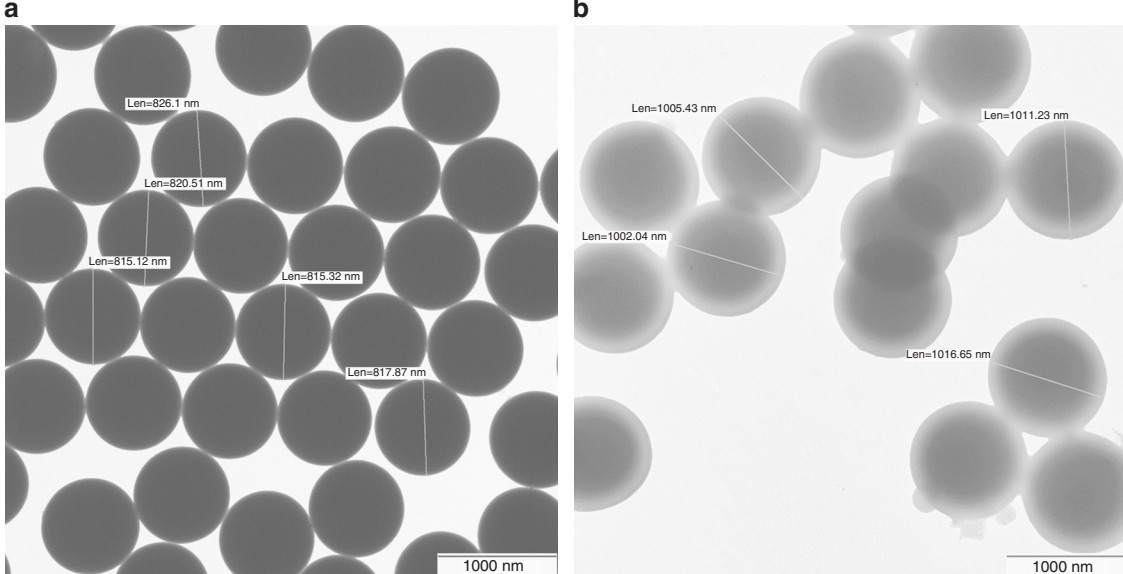

**Fig. 6** Electron microscopic images of targets: **a** polystyrene spheres and **b** hollow spheres made of polymethylmethacrylate

the center. In these regions the amplitude is set to zero. The remaining phase object is smaller and acts on the phase in a stronger manner than in the under-critical case. The plasma was considered as thin lens. The obtained phase and amplitude maps were used to solve the Kirchhoff's diffraction formula to predict the intensity distribution at the scatter screen. To mimic a flat-top beam we normalized the obtained diffraction patterns with respect to the undisturbed beam and applied an aperture.

**Wide angle spectrometer and image plate evaluation**. We employed a slit spectrometer in our experiment. Herby a magnetic field disperses different particle

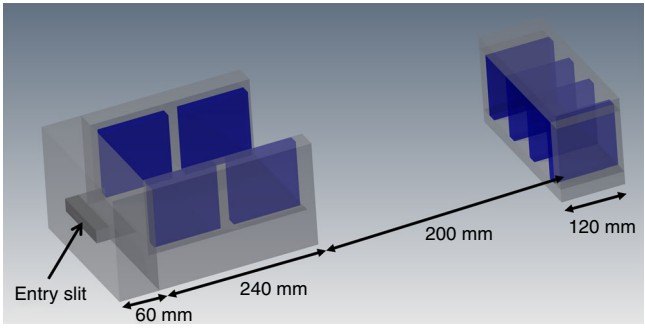

**Fig. 8** Ion spectrometer. 3D view of the employed particle spectrometer including relevant dimensions. The ions enter through a 0.5 mm high slit made out of tungsten into the first dipole magnet. After a drift length of 200 mm they enter the second dipole magnet. The detector was located directly at the exit of the second magnet

momenta spatially. Different particle species can have the same momentum. To obtain proton spectra we used degraders to prevent the carbon ions to reach the spectrometer (protons have a higher penetration depth than carbon ions with the same momentum). Heavy ion spectra (in our case carbon and oxygen) are sacrificed for spatial information. The magnetic slit spectrometer (Fig. 8) consists out of two dipole magnets. The first magnet was located 240 mm behind the target. The magnet had a length of 240 mm and a gap of 170 mm. The entry slit was made out of 20-mm-thick tungsten blocks on a 60-mm-thick steel front plate and had a width of 500 μm. Its orientation was parallel to the laser polarization. The distance between detector and slit amounted to 620 mm. A second dipole magnet was placed directly in front of the detector to increase the dispersion further. The second magnet consisted out of three gaps with 45 mm width and had a length of 120 mm. The magnetic field in the center of the first dipole magnet amounts to 0.1 and 0.3 T for the second. We employed 3D particle tracking to account for the inhomogeneous magnetic fields of the spectrometer. The necessary magnetic field maps were measured with a three-axis hall probe.

The evaluation routine for the proton spectra will be presented exemplarily on shoot #11. We used Bas-TR image plates (IP) as detectors covered in 100-μm-thick aluminum foil. Parts of the detector were additionally covered with 1-mm-thick CR39 nuclear track detectors wrapped in 15-μm-thick aluminum. Positions of CR39 are shown in Fig. 9b (yellow squares). The IPs were scanned with an MS-FLA5100 scanner from Fuji with a resolution of 100 μm. In the case of saturation (saturation only occurred for carbon data) the image plates were scanned twice. The obtained raw data were converted into photostimulated luminescence (PSL) using Fuji's conversion plugin for ImageJ. Scans were composed into an HDR image shown in Fig. 9a. A via particle tracing obtained energy-angle map was aligned within the PSL image. Some important iso-energy lines for protons are shown in Fig. 9a–c. Protons with energy of 3.4 MeV are able to penetrate the 100-μm-thick aluminum shielding of the IPs representing the low-energy cutoff of our spectrometer. Protons with 11 MeV are able to penetrate regions which were additionally covered with CR39. At regions where protons with an energy of 24.8 MeV would be situated C6+ ions with 75 MeV are able to penetrate the 100-μm-thick aluminum. C6+ ions with energies exceeding 249 MeV are be able to penetrate the 0.9-mm-thick CR39 detector together with 130 μm alumina. Protons with the same deflection correspond to an kinetic energy of 83 MeV. In no shoot we recorded image plate signal at these deflections. This implies that we were able

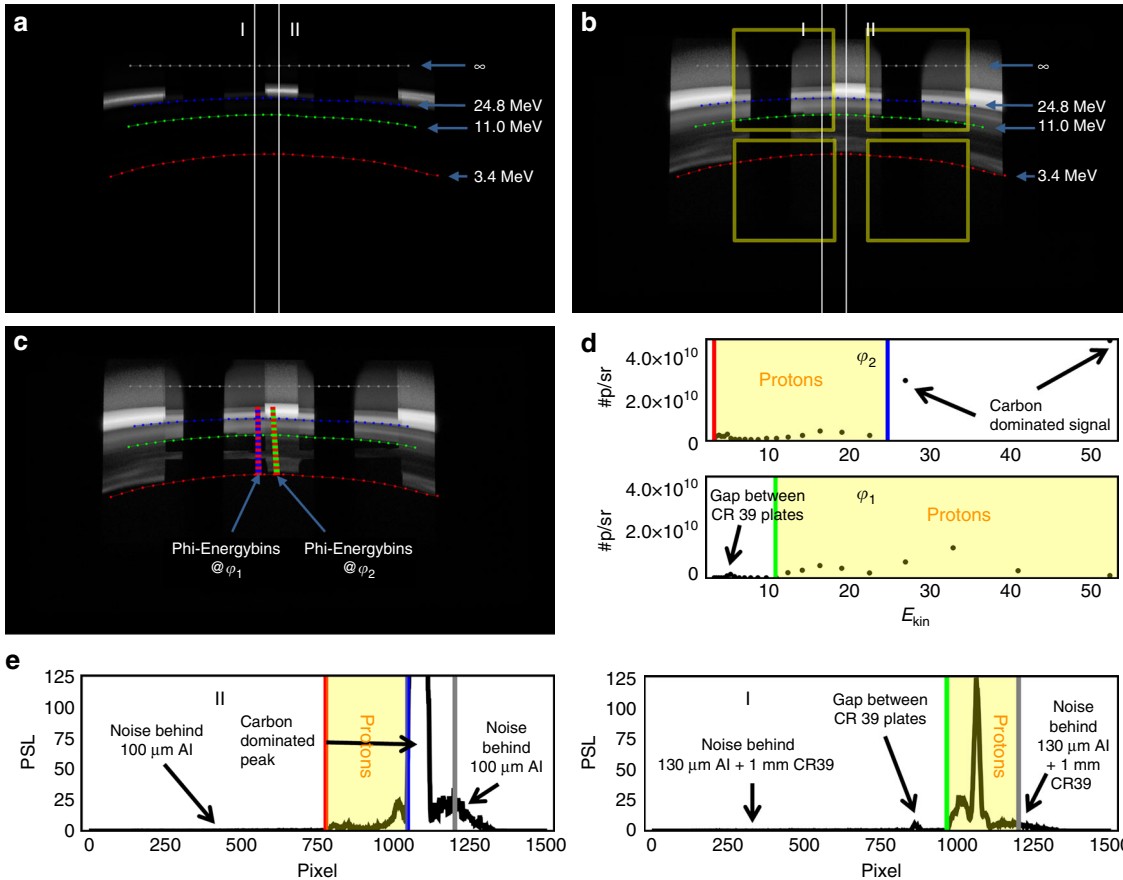

**Fig. 9** Image Plates data: **a** raw data with iso-energy lines for protons, **b** logarithmic representation of **a**, yellow squares indicate the position of CR39 plates, **c** energy-angle bins on the detector, **d** spectra obtained from **c**, **e** PSL raw data lineout

to discriminate uniquely proton signal from carbon signal for the proton energy range spanning from 3.4 to 83 MeV (yellow squares in Fig. 9d, e).

Lineouts of the PSL signal are shown in Fig. 9e. The left lineout shows the PSL signal which was shielded by 100 μm aluminum (Position II in Fig. 9a, b). The right lineout shows the PSL signal which was shielded by 130 μm aluminum and 0.9 mm CR39 (Position I in Fig. 9a, b). Due the different degraders regions of the IP can be identified that can only be reached by protons indicated as yellow regions in Fig. 9d, e. The cutoff lines in Fig. 9a–c are shown as vertical lines with corresponding color. We constructed energy-angle bins shown in Fig. 9c (blue, green, and red quadrilaterals). Herby the energy bin width has been chosen in such a way that corresponding bins in detector space had a height of 1500 μm, which corresponds to the slit projection on the IP. Due to the negligible angle dependence of the raw data signal, the angle bin width has been arbitrarily chosen to be 2792 mrad. Different angle bin sizes yielded similar results. One energy-angle bin contains about 450 pixel of the IP. Pixel which were partially covered by a bin were weighted according to their overlap. The conversion function from PSL to #p/pixel has been calibrated with a Tandem accelerator (MLL). The energy loss of the protons in the degraders has been taken into account for each bin individually. The final spectra are obtained at the borders of the CR39. Energy bins with values between 3.5 and 22 MeV were evaluated behind 100 μm aluminum (Fig. 9c red–green quadrilaterals). For energies above 22 MeV the IP signal behind CR39 was used (Fig. 9c red–blue quadrilaterals). The corresponding spectra are shown in Fig. 9d. Also here the cutoff lines are indicated by vertical lines with corresponding color. Yellow boxes mark regions where the signal is purely caused by protons. The resulting final spectrum is shown in Fig. 2c. Regarding carbon ion energies we can only make weak statements based on cutoff lines. In the experiment 100 μm aluminum was penetrated by carbon ions with a kinetic energy of at least 75 MeV. The absence of a cutoff line behind CR39 in our shoots evidences that the $C^{6+}$ energies were smaller than 250 MeV.

**Particle-in-cell simulations**. Particle-in-cell simulations were performed using a 0.2.0 pre-release of PIConGPU. The simulated 3D domain spans $104 \times 139 \times 104$ μm $(x, y, z)$ with $3456 \times 4624 \times 3456$ cells, resolving each $\Delta x = \Delta y = \Delta z = 30$ nm and $\Delta t = 57.5$ as. For the interaction with the main pulse, a pre-expanded, pre-ionized target with radial Gaussian density profile is modeled. Peak density corresponds to $n_e = 0.8\, n_c$ ($\omega_{pe} \cdot \Delta t < 0.09$) with a standard deviation of $\sigma = 3.0625$ μm. Its constituents $C_5O_2H_8$ are modeled after PMMA, assuming no de-mixing before the main interaction. GSI's PHELIX laser is modeled as finite Gaussian beam with central wavelength $\lambda = 1053$ nm, linear polarization in $x$ and propagation in $y$. Focusing the laser pulse on target, a normalized field strength $a_0 = 17$ in a 13.5 μm focal spot over a pulse length $\Delta t = 500$ fs (both FWHM intensity) interacts self-consistently with the initial density distribution. Self-consistent numerical methods deployed are the Yee field solver, trilinear field-to-particle interpolation, Boris particle push, Esirkepov current deposition, six initial macroparticles per cell with TSC shape. Consistently, boundary conditions are treated as absorbing and starting conditions as quasi-neutral.

The overall simulation run on 8000 Nvidia K20x GPUs on Titan (ORNL) for 42,000 time steps within 11.5 h. During runtime, 1 PByte of raw output was generated which was aggregated with ADIOS and on-the-fly compressed with zlib. Parallel post-processing was performed on the Rhea cluster (ORNL), accessing the same file system. Extensive preparatory 2D3V simulations were executed on the cluster Hypnos (HZDR) with 8-16 Nvidia K80 GPUs, scanning the pre-expansion of the target from full density down to 0.016 $n_c$ and variations in density profiles. Due to the symmetry assumptions of 2D simulations (infinite cylinders in the third dimension), these simulations were not conclusive and made 3D3V simulations necessary.

**Coulomb repulsion calculation**. We generated low resolution $(\Delta x = \Delta y = \Delta z = 500$ nm) 3D density maps for electrons carbons/oxygens and protons for $t = -260$ fs. The 3D maps were deduced from $xy$ and $yz$ density slices via interpolation. To account for neutralizing effects of bound electrons we reduced the carbon/oxygen charge density by the overlaying electron density. The so obtained 3D charge density distributions of carbons/oxygen and protons were used as input in a Coulomb repulsion code (General Particle Tracer (GPT)). Starting velocities were assumed to be monoenergetic. We used mean kinetic energies extracted of spectra from the simulation at $t = -260$ fs as starting parameters.

**Data availability**. The data that support the plots within this article and other findings of this study are available from the corresponding authors on reasonable request. Additionally, 3D3V particle-in-cell simulation input parameters and data of Fig. 3 are available under DOI:10.5281/zenodo.1005729.

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

## Acknowledgements

This work was supported by the DFG-Cluster of Excellence Munich Centre for Advanced Photonics and the SFB Transregio 18. This research used resources of the Oak Ridge Leadership Computing Facility located in the Oak Ridge National Laboratory, which is supported by the Office of Science of the Department of Energy under contract DE-AC05-00OR22725. This project has received funding from the European Unions Horizon 2020 research and innovation programme under grant agreement No 654220. A.H. received travel grants from the Nvidia GPU Center of Excellence Dresden. P.G.T and F.H.L. acknowledge funding by BMBF under contract 05P15WMEN9. J.G. acknowledges support from Hanns-Seidel-Stiftung. T.M.O. acknowledges support from IMPRS-APS. T.F.R acknowledges support from the German Academic Scholarship Foundation. M.G. and C.G.S. would like to thank the Fonds der Chemischen Industrie and the German Research Foundation (DFG GA 2169/5-1) for partial support of this work. This work has been carried out within the framework of the EUROfusion Consortium and has received funding, through the ToIFE, from the European Union's Horizon 2020 research and innovation programme under grant agreement number 633053. The authors acknowledge the excellent support of the PHELIX operating team throughout the two beam times. We acknowledge René Widera and all further contributors to the open-source code PIConGPU for enabling our simulations. We acknowledge Rafael Ramis for fruitful discussions on pre-plasma expansion.

## Author contributions

P.H. designed the experiments and analyzed the experimental data. J.S. provided guidance and supervision of the experimental team. P.H., T.M.O, J.G., and D.H. developed and optimized the Paul trap. P.H., T.M.O, V.B., B.B., J.G., D.H., J.H., F.H.L, P.N., P.G.T., T.F.R., F.W., and B.Z. participated in the experimental campaigns. J.S. and P.H. wrote the paper. M.G. and C.G.S. developed and provided the hollow sphere targets. A.H. performed and analyzed particle-in-cell simulations and developed required particle-in cell

methods. A.H. and T.K. designed simulation setups. P.H., A.H., and T.K. interpreted the simulation results. M.B. and U.S. provided guidance and supervision of the simulation and theory team.

## Additional information

**Competing interests:** The authors declare no competing financial interests.

