## [Peer Review File · Nature Communications]

Reviewers' comments:

Reviewer #1 (Remarks to the Author):

This manuscript reports the experimental results on reproducible production of laser-accelerated proton bunch with narrow energy spread (down to 25%) at the central energy of 20 – 40 MeV from a 1 micrometer diameter plastic sphere, the target positioning of which is implemented with a micrometer-precision Paul trap technique in the focus of the PHELIX laser at GSI delivering a 150 J, 500 fs laser pulse at 1054 nm wavelength with excellent temporal contrast ($\sim 10^{10}$ at ns ASE level) for preventing preplasma formation and destruction of the target prior to the main pulse interaction without the aid of plasma mirrors. In laser-driven ion, specifically proton, acceleration, a number of efforts for nearly the last two decades have been devoted to achieve the requirements of the proton radiation therapy demanding the maximum energy higher than 200 MeV and several 10¹⁰ protons/s with a 1%-level monoenergetic property. In this context, the present achievement would be one of the significant milestones toward proton radiation therapy, demonstrating a new target positioning method that provides a frameless target system with capabilities of diminishing background debris and repetitive shooting for laser-solid interaction experiments. Accordingly, this manuscript could be publishable in Nature Communications in viewpoint of novelty and significance in the field of high energy density physics with reservations about the following minor revision and presentation of more informative data that enhance author's conclusion:

1) Regarding recent progress on the maximum energy protons, authors should cite the paper [I. J. Kim et al., Phys. Plasmas 23, 070701 (2016)] reporting the result of 93 MeV protons via radiation pressure acceleration.

2) The details of the experimental setup too much simply illustrated in the inset of Fig. 1 without any explanation is not necessarily informative for the readers. Authors should show the detail of more informative drawing of the setup including the configuration of the diagnostics and the target system as well as the laser main and probe beams.

3) Regarding the differential proton spectra shown in Fig. 2 (a), authors should disclose more detailed information of the magnetic spectrometer (e.g. there is no information of the magnetic field and BL) and analysis of the imaging plate (IP) data as Supplementary Information. For instance, at a glance of Fig. 2 (a), the energy and spatial resolution of the spectrometer is not high-enough because one pixel of IP scanned image seems to correspond to ~ 5 MeV proton kinetic energy spread and angular spread of ~ 2 degrees (30 mrad). A monoenergetic peak at 30 MeV seems to consist of only 3 pixel IP data in Fig. 2 c). For convincing results, authors should show a scanned two dimensional raw image of IP over a large spatial and spectral area and its lineouts on energy and divergence axes in PSL value, indicating the background level (or detection limit) from x-rays and secondary emitted electrons.

4) While in Methods, it is mentioned that CR39 nuclear track detectors with 1 mm thickness are used for discrimination of ions and protons, there is no description on the contamination of carbon ions in main text. It would be very important to show the ion discrimination from proton for evidencing author's claim of finding the narrow energy spread protons with almost no low energetic background, because a charge discriminating spectrometer such as Thomson parabola is not used in this experiment.

5) Regarding directional angular distribution of proton emission, authors should show experimental evidence of large directionality such as measurement of the divergence angle of proton beam, which would be very important information not only for understanding acceleration mechanism and also for the practical application to proton therapy.

Reviewer #2 (Remarks to the Author):

Referee report

Isolated Proton Bunch Acceleration by a Peta-Watt Laser Pulse

By P. Hiltz et. al.

The article by Hiltz et. al. address an important problem in laser-target interaction that has been pursued for quite a while: generation of monoenergetic protons using high-power (PW) lasers. A number of schemes have been previously devised, but with limited success. The recent work not only demonstrates experimentally and theoretically the generation of monoenergetic proton bunches, but uses well-known facts to their advantage, such as (i) optimum energy absorption at near-critical density and (ii) the utilization of mass-limited targets. The work is original and on my opinion will contribute significantly to the advancement of this field. I recommend it for publication provisionally, after suggested minor improvements and addressing the issues listed below..

Major issues:

1. The advantage of mass-limited targets and their overall impact should be emphasized (T. Kluge, PoP 17, 123103 (2010), S. Buffechoux, PRL 105, 015005 (2010)), particularly because members of the team have studied it in the past and have first-hand knowledge.
2. Critical parameters of the interaction are not communicated, many of which are certainly available from PIC and perhaps from experiment. In particular,
 - what is the energy absorption by the sphere ?
 - what is the conversion efficiency of laser energy into protons (perhaps those with energy > 1 MeV)
 - what other ions are accelerated and what is their maximum energy ?
 - what is the laser energy on target
3. There is a mismatch between laser parameters: the laser energy on target is approximately $I \cdot D^2 \cdot \tau$, which for $I=7 \times 10^{20}$, $\tau=0.5$ ps and $D=3.7$ microns yield 50 J, not 150 J. I am aware of the widely used concept e.g. "30 % of the laser energy is in the focal spot", but even for a Gaussian one gets 27 J in the focal spot and another 27 J in the wings.

More specifically, 10^{10} protons (page 8) times 30 MeV (average energy, Figure 2c) yields only 0.05 J of energy into protons. This is barely 0.03 % ! Why would a PW laser have such low conversion efficiency ?

What is the price paid for hitting a single sphere in terms of energy absorption ? What fraction of the laser energy is actually absorbed by the blob of plasma ?

Minor issues:

1. page 4, middle: "...our measured values are 30 to 100 times larger". Not clear. I believe that what you have in mind is that $dN/d\Omega$ is 30 to 100 times larger because the proton beam is directional.
2. page 6, middle: "As protons are the species of highest mobility they are ...". I would recommend "...As protons are the species with largest q/M they are ...".
3. page 6, middle: "It represents the accumulatively...". I would recommend "It represents the cumulative..."
4. page 6, middle: "...i.e. laser energy...". I believe it is laser fluence, since $\int I(t) dt$ is fluence.

We want to thank the referees for their valuable suggestions regarding our manuscript, which with no doubt constructively improved the manuscript. Below we respond to the specific comments point by point.

Reviewers' comments:

Reviewer #1 (Remarks to the Author):

This manuscript reports the experimental results on reproducible production of laser-accelerated proton bunch with narrow energy spread (down to 25%) at the central energy of 20 – 40 MeV from a 1 micrometer diameter plastic sphere, the target positioning of which is implemented with a micrometer-precision Paul trap technique in the focus of the PHELIX laser at GSI delivering a 150 J, 500 fs laser pulse at 1054 nm wavelength with excellent temporal contrast ($\sim 10^{10}$ at ns ASE level) for preventing preplasma formation and destruction of the target prior to the main pulse interaction without the aid of plasma mirrors. In laser-driven ion, specifically proton, acceleration, a number of efforts for nearly the last two decades have been devoted to achieve the requirements of the proton radiation therapy demanding the maximum energy higher than 200 MeV and several 10¹⁰ protons/s with a 1%-level monoenergetic property. In this context, the present achievement would be one of the significant milestones toward proton radiation therapy, demonstrating a new target positioning method that provides a frameless target system with capabilities of diminishing background debris and repetitive shooting for laser-solid interaction experiments. Accordingly, this manuscript could be publishable in Nature Communications in viewpoint of novelty and significance in the field of high energy density physics with reservations about the following minor revision and presentation of more informative data that enhance author's conclusion:

1) Regarding recent progress on the maximum energy protons, authors should cite the paper [I. J. Kim et al., Phys. Plasmas 23, 070701 (2016)] reporting the result of 93 MeV protons via radiation pressure acceleration.

Thank you for spotting this lack. Clearly the reference should be included which we now did.

We changed:

“... exceeding 85 MeV have been recently demonstrated⁴.”

To:

“exceeding 85 MeV have been recently demonstrated in independent experiments^{4,5}”

2) The details of the experimental setup too much simply illustrated in the inset of Fig. 1 without

any explanation is not necessarily informative for the readers. Authors should show the detail of more informative drawing of the setup including the configuration of the diagnostics and the target system as well as the laser main and probe beams.

We agree with the referee that the setup schematic in Fig 1 was indeed oversimplified.

We changed the inset from:

“

“

To:

“

”

For interested readers we added a setup figure including more technical details of the experiment into the supplementaries.

The supplementary information was extended by:

“

Detailed Experimental Setup:

Fig4: Schematic overview of the experimental setup. 1) incoming Phelix laser beam 2) Paul trap power supply 2a) 660 nm illumination laser 2b) electro-optical diagnostic for target damping and positioning 3) Paul trap electrodes 4) trapped target 5) scatter screen for transmitted light with 5a) and without 5b) target, 6) magnetic spectrometer, 7) IP raw proton/ion data (without degraders)

Fig.4 shows schematically our experimental setup. All numbered components are described in more detail in the following sections.

“

With the revised inset in Fig 1. we hope to preserve the continuity of the main text on the one hand. On the other hand we extended the supplementaries extensively for technically interested readers (see also our replies to questions 3. and 4.).

3)Regarding the differential proton spectra shown in Fig. 2 (a), authors should disclose more detailed information of the magnetic spectrometer (e.g. there is no information of the magnetic field and BL) and analysis of the imaging plate (IP) data as Supplementary Information. For instance, at a glance of Fig. 2 (a), the energy and spatial resolution of the spectrometer is not high-enough because one pixel of IP scanned image seems to correspond to ~5 MeV proton kinetic energy spread and angular spread of ~2 degrees (30 mrad). A monoenergetic peak at 30 MeV seems to consist of only 3 pixel IP data in Fig. 2 c). For convincing results, authors should show a scanned two dimensional raw image of IP over a large spatial and spectral area and its lineouts on energy and divergence axes in PSL value, indicating the background level (or detection limit) from x-rays and secondary emitted electrons.

The spectrometer resolution is limited by the projected slit width of the spectrometer entrance (500 μm Slit projected to 1500 μm). The image plate readout resolution is 100 μm . The 30 MeV band spreads over 4.5 mm on the IP, which corresponds to the 3 energy bins shown in Fig 2c.

In conjunction with the extended setup figure, we extended the information on our spectrometer in the supplementary information. The requested two dimensional raw image of the IP and line outs with indicated background level are now shown .

The supplementary information on the particle spectrometer now reads:

“

Wide Angle Spectrometer and Image Plate Evaluation.

Fig. 8: Ion Spectrometer

We employed a slit spectrometer in our experiment. Herby a magnetic field disperses different particle momenta spatially. Different particle species can have the same momentum. To obtain proton spectra we used degraders to prevent the carbon ions to reach the spectrometer (Protons have a higher penetration depth than carbon ions with the same momentum). Heavy ion spectra (in our case carbon and oxygen) are sacrificed for spatial information. The magnetic slit spectrometer (Fig. 8) consists out of two dipole magnets. The first magnet was located 240 mm behind the target. The magnet had a length of 240 mm and a gap of 170 mm. The entry slit was made out of 20 mm thick tungsten blocks on a 60 mm thick steel front plate and had a width of 500 μm . Its orientation

was parallel to the laser polarization. The distance between detector and slit amounted to 620 mm. A second dipole magnet was placed directly in front of the detector to increase the dispersion further. The second magnet consisted out of three gaps with 45 mm width and had a length of 120 mm. The magnetic field in the center of the first dipole magnet amounts to 0.1 T and 0.3 T for the second. We employed 3D particle tracking to account for the inhomogeneous magnetic fields of the spectrometer. The necessary magnetic field maps were measured with a three axis hall probe.

Fig 9 Image Plates data: a) PSL raw data with iso-energy lines for protons, b) logarithmic representation of a) ,yellow squares indicate the position of CR 39 plates c) energy-angle bins on the detector d) spectra obtained from c) e) PSL raw data lineout.

The evaluation routine for the proton spectra will be presented exemplarily on shoot #11. We used Bas-TR image-plates (IP) as detectors covered in 100 μm thick aluminum foil. Parts of the detector were additionally covered with 1 mm thick CR39 nuclear track detectors wrapped in 15 μm thick

aluminum. Positions of CR 39 are shown in Fig. 9b (yellow squares). The IPs were scanned with an MS-FLA5100 scanner from Fuji with a resolution of 100 μm . In the case of saturation (saturation only occurred for carbon data) the image plates were scanned twice. The obtained raw data was converted into PSL using Fuji's conversion plugin for ImageJ. Scans were composed into an HDR image shown in Fig. 9a. A via particle tracing obtained energy-angle map was aligned within the PSL image. Some important iso- energy lines for protons are shown in Fig. 9a,b,c. Protons with energy of 3.4 MeV are able to penetrate the 100 μm thick aluminum shielding of the IP's representing the low energy cut off of our spectrometer. Protons with 11 MeV are able to penetrate regions which were additionally covered with CR 39. At regions where protons with an energy of 24.8 MeV would be situated C6+ ions with 75 MeV are able to penetrate the 100 μm thick aluminum. C6+ ions with energies exceeding 249 MeV are able to penetrate the 0.9 mm thick CR 39 detector together with 130 μm alumina. Protons with the same deflection correspond to a kinetic energy of 83 MeV. In no shoot we recorded image plate signal at these deflections. This implies that we were able to discriminate uniquely proton signal from carbon signal for the proton energy range spanning from 3.4 to 83 MeV (yellow squares in Fig. 9d,e).

Lineouts of the PSL signal are shown in Fig. 9e. The left lineout shows the PSL signal which was shielded by 100 μm aluminium (Position II in Fig. 9a,b). The right lineout shows the PSL signal which was shielded by 130 μm aluminum and 0.9 mm CR39 (Position I in Fig. 9a,b). Due to the different degraders regions of the IP can be identified that can only be reached by protons indicated as yellow regions in Fig. 9d,e). The cut off lines in Fig. 9a-c are shown as vertical lines with corresponding color. We constructed energy-angle bins shown in Fig. 9c (blue, green and red quadrilaterals). Herby the energy bin width has been chosen in such a way that corresponding bins in detector space had a height of 1500 μm , which corresponds to the slit projection on the IP. Due to the negligible angle dependence of the raw data signal, the angle bin width has been arbitrarily chosen to be 2,792 mrad. Different angle bin sizes yielded similar results. One energy-angle bin contains about 450 pixel

of the IP. Pixel which were partially covered by a bin were weighted according to their overlap. The conversion function from PSL to #p/pixel has been calibrated with a Tandem accelerator (MLL). The energy loss of the protons in the degraders has been taken into account for each bin individually. The final spectra are obtained at the borders of the CR39. Energy bins with values between 3.5 and 22 MeV were evaluated behind 100 μm aluminum (Fig. 9c red-green quadrilaterals). For energies above 22 MeV the IP signal behind CR39 was used (Fig. 9c red-blue quadrilaterals). The corresponding spectra are shown in Fig. 9d. Also here the cut off lines are indicated by vertical lines with corresponding color. Yellow boxes mark regions where the signal is purely caused by protons. The resulting final spectrum is shown in Fig. 2c. Regarding carbon ion energies we can only make weak statements based on cutoff lines. In the experiment 100 μm aluminum was penetrated by carbon ions with a kinetic energy of at least 75 MeV. The absence of a cutoff line behind CR 39 in our shoots evidences that the C6+ energies were smaller than 250 MeV.

”

4)While in Methods, it is mentioned that CR39 nuclear track detectors with 1 mm thickness are used for discrimination of ions and protons, there is no description on the contamination of carbon ions in main text. It would be very important to show the ion discrimination from proton for evidencing author’s claim of finding the narrow energy spread protons with almost no low energetic background, because a charge discriminating spectrometer such as Thomson parabola is not used in this experiment.

The reviewer raises a clear point on insufficient documentation of the particle diagnostics. The lack of this information on the spectrometer caused some misunderstanding which we regret.

In accordance with point 2 and 3, we extended the supplementary materials, on the spectrometer extensively including raw data PSL Values etc. and evaluation routines. All raised points should be hereby clarified immediately.

In particular, the discrimination from carbon ions and protons was achieved by using Al and CR 39 as degraders. We only evaluated image plate sections which could only be reached by protons.

5)Regarding directional angular distribution of proton emission, authors should show experimental evidence of large directionality such as measurement of the divergence angle of

proton beam, which would be very important information not only for understanding acceleration mechanism and also for the practical application to proton therapy.

Unfortunately we do not have a direct measurement of the proton beam divergence. Instead we extracted the directionality from logical considerations from absolute particle numbers as presented in the text. The number of protons emitted into a solid angle, as shown in Fig 2b, is 30...100 times larger than one could expect if all protons in the target were emitted into 4π . Therefore, we estimate our opening angle to smaller than $4\pi/100$... $4\pi/30$ which we call directed. The such estimated divergence angle agrees also well with divergence angle observed in the simulation presented Fig. 3f. In this sense, the divergence angle is not smaller, but also not larger than observed in (most) foil target experiments.

Concluding remark:

We thank referee 1 for the feedback and agree on the spotted insufficiencies in the technical depth of our description. Hence we extended the supplementary information slightly for clarification. We added an image of the Paul trap. A short description of the hollow spheres and a schematic image for the diffraction calculations were added.

Reviewer #2 (Remarks to the Author):

Referee report

Isolated Proton Bunch Acceleration by a Peta-Watt Laser Pulse

By P. Hiltz et. al.

The article by Hiltz et. al. address an important problem in laser-target interaction that has been pursued for quite a while: generation of monoenergetic protons using high-power (PW) lasers. A number of schemes have been previously devised, but with limited success. The resent work not only demonstrates experimentally and theoretically the generation of monoenergetic proton bunches, but uses well-known facts to their advantage, such as (i) optimum energy absorption at near-critical density and (ii) the utilization of mass-limited targets. The work is original and on my opinion will contribute significantly to the advancement of this field. I recommend it for publication provisionally, after suggested minor improvements and addressing the issues listed below..

Major issues:

1. The advantage of mass-limited targets and their overall impact should be emphasized (T. Kluge, PoP 17, 123103 (2010), S. Buffechoux, PRL 105, 015005 (2010)), particularly because members of

the team have studied it in the past and have first-hand knowledge.

We agree and regret not having included these references in the first place. We now cite them as suggested by saying:

“

Spatially limited targets, so called mass limited targets, have attracted severe interest in theory¹⁵ and experiment^{9,16,17}

”

**2. Critical parameters of the interaction are not communicated, many of which are certainly available from PIC and perhaps from experiment. In particular,
- what is the energy absorption by the sphere ?:**

The referee raises an interesting question regarding laser absorption encountered in the experiment. Unfortunately this question is quite complicated to answer in our case due to the smooth transition from organized acceleration to Coulomb repulsion dominated regime. In the first stage a significant amount of energy is stored into quasi-static electric fields, which due to the spherical geometry in combination with diffraction effects can not be separated from the laser fields in the simulation in a uniquely manner. At later stages electrons undergo strong vacuum acceleration (up to 100-200 MeV). The hereby absorbed laser energy does not contribute to proton or ion acceleration. We are motivated by Coulomb repulsion calculations that after -260 fs the ongoing laser irradiation becomes irrelevant. Ergo the presented experiment could also have been conducted with a much smaller laser system, hereby demonstrating great future potential. To answer the question, we have argued that in the current setting only up to 8 Joule (namely the power integrated up to t=-260 fs) contribute to the ion acceleration process. How much of this energy is absorbed by the sphere is not immediately evident.

- what is the conversion efficiency of laser energy into protons (perhaps those with energy > 1 MeV):

- what other ions are accelerated and what is their maximum energy ?:

- what is the laser energy on target:

We are thankful for the referees hint to improve the discussion. We emphasised the suggested aspects on conversion efficiencies and energy on target by changing:

“

...as long as possible. Future studies, both theoretical and experimental, should therefore concentrate on spectral, temporal, polarisation and spatial shaping of the laser pulse to further optimize the electron dynamics that is responsible for the generated acceleration fields. On this track...

”

into:

“

as long as possible. In this context it is instructive to examine the conversion efficiency from laser energy into kinetic energy of protons/ions. In our simulation 72 J laser energy pass through the plasma. The accumulated final kinetic energy of all protons is 181 mJ, 196 mJ for oxygen ions and 367 mJ for carbon ions, respectively. This leads to an overall energy conversion efficiency into protons of 0.25% and 1% into protons and ions combined. Until -260 fs only 7.9 J have passed the plasma, the remaining part of the laser pulse does not contribute to proton/ion acceleration. The effective laser energy conversion efficiency may therefore be arguably ~10x larger: 2.6% into protons and 11% for protons and ions combined. Consequently, future studies, both theoretical and experimental, should concentrate on spectral, temporal, polarisation and spatial shaping of the laser pulse to further optimize the electron dynamics that is responsible for the generated acceleration fields in the first acceleration stage. On this track, target

”

Regarding carbon ions we added the following statement into the supplementary information:

“

Regarding carbon ion energies we can only make weak statements based on cutoff lines. In the experiment 100 μm aluminum was penetrated by carbon ions with a kinetic energy of at least 75 MeV. The absence of a cutoff line behind CR 39 in our shoots evidences that the C6+ energies were smaller than 250 MeV.

”

3. There is a mismatch between laser parameters: the laser energy on target is approximately $I \cdot D^2 \cdot \tau$, which for $I=7 \times 10^{20}$, $\tau=0.5$ ps and $D=3.7$ microns yield 50 J, not 150 J. I am aware of the widely used concept e.g. "30 % of the laser energy is in the focal spot", but even for a Gaussian one gets 27 J in the focal spot and another 27 J in the wings.

Indeed, we found our formulation to be misleading on revision. The quoted 7×10^{20} W/cm² refer to the intensity on target which was positioned 1 - 1.5 Rayleigh-lengths out of focus to minimize pointing jitter effects.

We changed:

“...where the on-target peak laser intensity amounted to 7×10^{20} W/cm².”

To:

“...where the on-target peak laser intensity amounted up to 7×10^{20} W/cm² (The peak intensity in focus would have amounted to 2×10^{21} W/cm²).”

More specifically, 10^{10} protons (page 8) times 30 MeV (average energy, Figure 2c) yields only 0.05 J of energy into protons. This is barely 0.03 % ! Why would a PW laser have such low conversion efficiency ?

The referees finding is correct. The overall conversion efficiency is indeed very poor in our experiment. We have hopefully clarified this point with the detailed explanation above and the added explanation in the manuscript.

What is the price paid for hitting a single sphere in terms of energy absorption ? What fraction of the laser energy is actually absorbed by the blob of plasma ?

As the referee pointed out, the fractional laser energy absorbed by the plasma is indeed very small, and this point is explained now. Hence, in this demonstration experiment we indeed paid a large price. But the study shows for the first time a path towards optimization, and as explained, with the effectively used ~ 8 J of laser energy and actually comparably small intensity up to this point along the interaction, the demonstrated scheme promises an interesting investigative path over the next few years.

Minor issues:

We want to thank the referee for spotting these obvious mistakes.

1. page 4, middle: "...our measured values are 30 to 100 times larger". Not clear. I believe that what you have in mind is that $dN/d\Omega$ is 30 to 100 times larger because the proton beam is directional.

Changed from:

“Compared to a scenario of an ideal Coulomb explosion, where all protons in the target are emitted into 4π sr (dashed red line in Fig. 2b), our measured values are 30 to 100 times larger. Though not visible in the small angular range of our particle spectrometer, this comparison evidences a large degree of directionality of the accelerated proton bunch.”

To:

“Contrary to foil and mechanically mounted targets, the initial number of protons is known in our case. Herby we can deduce the #p/sr for an ideal Coulomb explosion, where all protons in the target are emitted into 4π sr (dashed red line in Fig. 2b). Our measured values for #p/sr are 30 to 100 times larger compared to the isotropic ideal Coulomb explosion. Though not visible in the small angular range of our particle spectrometer, this comparison evidences a large degree of directionality of the accelerated proton bunch, also in accordance with our simulations.”

2. page 6, middle: "As protons are the species of highest mobility they are ...". I would recommend "...As protons are the species with largest q/M they are ...".

Changed according to your suggestion into:

“...As protons are the ion species with largest charge to mass ratio they are...”

3. page 6, middle: "It represents the accumulatively...". I would recommend "It represents the cumulative..."

Changed as requested.

4. page 6, middle: "...i.e. laser energy...". I believe it is laser fluence, since $\int I(t) dt$ is fluence.

We apologize for the mistake. We in fact plot $\int I(t,r) dt d\sigma$ as indicated in figure 3a, that is indeed the energy which has passed the plasma.

We corrected the wording to “It represents the cumulatively time and spatially integrated laser intensity ...”

REVIEWERS' COMMENTS:

Reviewer #1 (Remarks to the Author):

The resubmitted manuscript reports on the production of laser-driven proton beams with narrow energy spread and directionality in a practically useful energy range for the first time. This revised manuscript and author's response to the reviewer's comments have been substantially improved and satisfactorily addressed, showing more informative presentation of the technical aspect and convincing descriptions on the data analysis in the supplementary information, which were quite insufficient in the previous manuscript. Accordingly, this manuscript is willingly recommended to be publishable in Nature Communications. As an optional revision, since a key point of this paper is a novel target system using ion trapping and optical tweezer technologies, the setup view shown in Fig. 4 may be illustrated in the main text preceding Fig. 1. For this change, the inset figure in Fig. 1 (a) could be eliminated. As a minor revision, in the sentence "Herby the energy bin width has been chosen in such a way that corresponding bins in detector space had a height of 1500 μm , ...", "Herby" should be "Hereby".

Reviewer #2 (Remarks to the Author):

The authors have responded to my comments satisfactory and made the suggested amends in the paper. The article content is physically sound, advances the field of laser-target interaction significantly meeting the criteria for publications by the Journal and contains sufficient novelty to be published in Nature Communications.

We want to thank the referees for their valuable suggestions regarding our manuscript, which with no doubt constructively improved the manuscript. Below we respond to the specific comments point by point.

REVIEWERS' COMMENTS:

Reviewer #1 (Remarks to the Author):

The resubmitted manuscript reports on the production of laser-driven proton beams with narrow energy spread and directionality in a practically useful energy range for the first time. This revised manuscript and author's response to the reviewer's comments have been substantially improved and satisfactorily addressed, showing more informative presentation of the technical aspect and convincing descriptions on the data analysis in the supplementary information, which were quite insufficient in the previous manuscript. Accordingly, this manuscript is willingly recommended to be publishable in Nature Communications. As an optional revision, since a key point of this paper is a novel target system using ion trapping and optical tweezer technologies, the setup view shown in Fig. 4 may be illustrated in the main text preceding Fig. 1. For this change, the inset figure in Fig. 1 (a) could be eliminated. As a minor revision, in the sentence "Herby the energy bin width has been chosen in such a way that corresponding bins in detector space had a height of 1500 μm ,", "Herby" should be "Hereby".

We understand the motivation of the reviewer#1 for the optional revision. Unfortunately we decided against it. Even if our target system is cutting edge technology we wanted to keep the focus on the physical findings rather than necessary technology. We wrote a paper concentrating on the target system itself, which got accepted in Review of Scientific Instruments a few day ago.

Concerning the minor revision we changed:

"...Herby..."

To:

"...Herbey..."

Reviewer #2 (Remarks to the Author):

The authors have responded to my comments satisfactory and made the suggested amends in the paper. The article content is physically sound, advances the field of laser-target interaction significantly meeting the criteria for publications by the Journal and contains sufficient novelty to be published in Nature Communications.